# Dynamics of SARS-CoV-2 IgG in Nursing Home Residents in Belgium Throughout Three BNT162b2 Vaccination Rounds: 19-Month Follow-Up

**DOI:** 10.3390/vaccines13040409

**Published:** 2025-04-15

**Authors:** Eline Meyers, Liselore De Rop, Claudia Gioveni, Fien Engels, Anja Coen, Tine De Burghgraeve, Marina Digregorio, Pauline Van Ngoc, Nele De Clercq, Laëtitia Buret, Samuel Coenen, Elizaveta Padalko, Els Duysburgh, Beatrice Scholtes, Jan Y. Verbakel, Stefan Heytens, Piet Cools

**Affiliations:** 1Department of Diagnostic Sciences, Faculty of Medicine and Health Sciences, Ghent University, 9000 Ghent, Belgium; eline.meyers@ugent.be (E.M.);; 2LUHTAR—Leuven Unit for HTA Research, Department of Public Health and Primary Care, KU Leuven, 3000 Leuven, Belgium; liselore.derop@kuleuven.be (L.D.R.); tine.deburghgraeve@kuleuven.be (T.D.B.); jan.verbakel@kuleuven.be (J.Y.V.); 3Department of Public Health and Primary Care, Faculty of Medicine and Health Sciences, Ghent University, 9000 Ghent, Belgium; 4Research Unit of Primary Care and Health, Department of General Medicine, Faculty of Medicine, University of Liège, 4000 Liège, Belgiumbeatrice.scholtes@uliege.be (B.S.); 5Department of Family Medicine & Population Health, Faculty of Medicine and Health Sciences, University of Antwerp, 2000 Antwerp, Belgium; samuel.coenen@uantwerpen.be; 6Laboratory of Medical Microbiology, Ghent University Hospital, 9000 Ghent, Belgium; 7Department of Epidemiology and Public Health, Sciensano, 1000 Brussels, Belgium

**Keywords:** COVID-19 vaccination, nursing home residents, booster, antibody dynamics

## Abstract

Background/Objectives: This study mapped antibody dynamics across three COVID-19 vaccination rounds (primary course, first, and second booster with BNT162b2) in Belgian nursing home residents (NHRs). Methods: Within a national SARS-CoV-2 serosurveillance study (February 2021–September 2022) across Belgian nursing homes, dried blood spots were collected, on which anti-spike SARS-CoV-2 IgG antibodies were quantified by ELISA in international units/mL (IU/mL). Sociodemographic data were collected at the study start and infection history and vaccination data at each sampling round. Results: Infection-naïve NHRs had low antibody levels after primary course vaccination (geometric mean concentration (GMC) 292 IU/mL, 95% confidence interval (95% CI): 197–432), but increased tenfold after first booster (GMC 2168 IU/mL, 95% CI: 1554–3027). While antibodies among NHRs significantly declined within six months after primary vaccination (*p* < 0.0001), they remained stable for nine months post-booster (*p* > 0.05). Among primary vaccine non-responders, 92% (95% CI: 82–97%) developed antibodies after the first booster (GMC 594 IU/mL, 95% CI: 416–849), though tenfold lower than initial responders (GMC 4642 IU/mL, 95% CI: 3577–6022). Conclusions: These findings demonstrate that NHRs require tailored vaccination, prioritizing repeated immunization to improve serological outcomes in poor responders such as infection-naive NHRs. Regular immune monitoring could aid in implementing evidence-based vaccine strategies, ensuring optimal protection for vulnerable populations against SARS-CoV-2 and other infectious threats.

## 1. Background

The coronavirus disease (COVID-19) pandemic, caused by the severe acute respiratory syndrome coronavirus 2 (SARS-CoV-2), posed a significant global health challenge, with older adults, particularly nursing home residents (NHRs), being among the most vulnerable and affected populations. In nursing home settings, NHRs often experienced higher morbidity and mortality rates during the pandemic due to older age, comorbidities, frailty, and close living conditions that facilitate viral transmission [1,2]. As such, nursing homes have been focal points in COVID-19 vaccination campaigns. In Belgium, NHRs were prioritized during the COVID-19 vaccination campaign, receiving primary course vaccination with BNT162b2 in a two-dose regimen (3-week interval, 30 μg) starting January 2021 [3]. Previous studies in Belgium and other countries demonstrated that antibody concentrations significantly wane in NHRs in the months following BNT162b2 primary course vaccination, sometimes dropping below detectable levels [4,5,6,7]. Moreover, a proportion of NHRs failed to mount a proper antibody response after vaccination, classifying them as poor or non-responders [4,8]. This is likely attributed to immunosenescence, which is an age-associated decline in the immune system [9]. In response to this observed waning humoral immunity after primary course vaccination, as demonstrated by our previous work [10], Belgian NHRs were administered a monovalent BNT162b2 booster dose (30 μg) in October–November of 2021. Moreover, in the following year, second BNT162b2 booster doses were introduced among Belgian NHRs. However, regional differences in vaccine rollout were present. In Flanders, second boosters (monovalent BNT162b2 vaccine, 30 μg) were administered in June 2022, while in Brussels and Wallonia, second boosters were administered later on, starting from October 2022, when the bivalent BNT162b2 vaccine was available [11,12,13].

The COVID-19 pandemic created unique settings due to changing epidemiological situations and vaccine approaches. Studies investigating antibody responses in NHRs to different COVID-19 vaccine regimens and booster doses exist, yet longitudinal follow-up of antibody responses in an NHR cohort throughout multiple vaccination rounds is rare. In this publication, we mapped the dynamics of SARS-CoV-2 antibodies among NHRs throughout three vaccination rounds in Belgium. Hereby, we focused on the following objectives. First, we aimed to compare the antibody response after primary course vaccination with the one after first booster vaccination. Secondly, we aimed to investigate whether NHR non-responders after primary course vaccination presented an antibody response after booster vaccination and if so, whether they reached comparable levels to initial responders. Thirdly, as a secondary objective, we aimed to study the effect of the second booster dose on antibody levels among NHRs in Flanders and compared it to antibody levels among NHRs in Brussels and Wallonia, where the second booster was administered later.

## 2. Materials and Methods

### 2.1. Study Design, Population, and Sample Size

The data in this publication were collected as part of a national SARS-CoV-2 serosurveillance study in Belgian nursing homes (SCOPE study). The first part of the SCOPE study ran from February 2021 to December 2021 with bimonthly sampling and included 1640 NHRs from 69 nursing homes across Belgium (SCOPE1) [1]. The second part of the study comprised an extended follow-up of 492 NHRs from 30 nursing homes across Belgium from December 2021 to September 2022 with sampling every three months (SCOPE2). Different subsets of NHRs from SCOPE1 and SCOPE2 were selected for the analyses presented in the current publication. An overview of the subsets and sampling timepoints is given in Figure 1.

Subset I included 200 NHRs randomly selected (random sample) within the SCOPE2 subset (on the condition that they received primary course vaccination by April 2021). Subset I was used to obtain a general overview of antibody dynamics in Belgian NHRs throughout follow-up, regardless of effects of vaccination rounds or natural infections. Additionally, subset I was used to longitudinally compare anti-SARS-CoV-2 spike protein S1 receptor-binding domain (S1RBD IgG) antibody concentrations after primary course vaccination with antibody concentrations after booster vaccination in infection-naive and infection-primed NHRs. Infection-naive NHRs are those with no prior confirmed SARS-CoV-2 infection, whereas infection-primed NHRs had confirmed infection prior to vaccination. To compare antibody levels after primary course vaccination and booster, subjects with a breakthrough infection, those who received a first booster before sampling in October 2021 or those who received a second booster during follow-up were excluded after the respective events. Subset II included 121 NHRs (61 non-responders and 60 matched responders) and was used to compare S1RBD IgG levels after first booster vaccination between non-responders and responders to primary course vaccination. Non-responders were defined when a subject’s first and/or second SARS-CoV-2 antibody rapid test result ≥ 14 days following primary course vaccination was negative. Responder controls were searched within the SCOPE1 subset matched for participant type, history of infection, and vaccination status of non-responders in a 1:1 ratio. Subjects with a breakthrough infection during follow-up were not selected. Subset III included 411 NHRs selected from the SCOPE2 study and was used to assess differences in S1RBD IgG levels between those who received a second booster and those who did not. Subjects who received a second booster dose before sampling in June 2022 and those who experienced a breakthrough infection during follow-up were excluded. More details on the subset selection can be found in the Appendix A. For subset I, sample size was chosen arbitrarily while for subset II and III, sample size was limited by the enrollment in the SCOPE study and the study selection criteria. A post hoc sample size calculation was performed and showed an achieved power of 100% for all subsets for a repeated-measures ANOVA with within–between interaction, assuming an effect size of 0.25 (medium effect) and an alpha of 0.05 (G Power Version 3.1) [14].

**Figure 1 vaccines-13-00409-f001:**
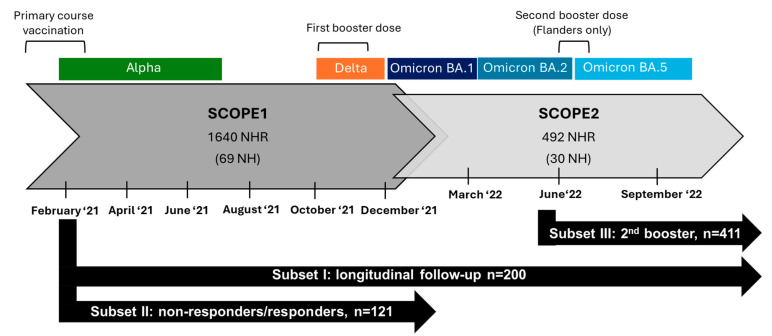
Overview of the subsets (black arrows) presented in this publication within a national SARS-CoV-2 serosurveillance study in Belgian nursing homes (grey arrows). Sampling times in the study, epidemiological waves in Belgium with respective dominant variant of concern [15,16], and timing of vaccination campaigns in nursing homes are presented. NHR: nursing home residents, NH: nursing homes.

### 2.2. Ethics

The study protocol received approval from the Ethics Committee of Ghent University Hospital (reference number BC-08719) and was conducted following the ethical principles of the Declaration of Helsinki. Informed consent was obtained from all participants or their legal representatives after explaining the study’s aims and procedures.

### 2.3. Data Collection

#### 2.3.1. Clinical Sample Collection

In order to assess SARS-CoV-2 IgG antibody concentrations, dried blood spot (DBS) samples were collected. Capillary blood was collected through a fingerprick using an 18G lancet with a depth of 1.8 mm (SARSTEDT Ag & Co., Nümbrecht, Germany), and transferred onto dried blood spot cards (EUROIMMUN, Lübeck, Germany). This procedure was carried out by trained personnel, ensuring that at least four 6 mm diameter circles on the card were saturated with blood and allowed to air dry for at least one hour. The samples were stored at −20 °C until further analysis [17].

#### 2.3.2. Questionnaires

Sociodemographic and clinical data were gathered through online questionnaires (LimeSurvey version 3.22, LimeSurvey GmbH, Hamburg, Germany), filled out by the nursing homes’ head nurses, based on the NHR’s medical file. Collected data included participant characteristics such as age, sex, and care dependency level (as according to the Katz evaluation scale, with category O, A, and B classified as low care-dependent and category C, Cd, and D as high care-dependent) [18]. Moreover, during each round of sample collection, information concerning the COVID-19 vaccination history (dose count, dates, and vaccine brand) and infection history (verified by PCR or antigen testing, or CT scan, along with test results and testing dates for positive cases) of NHRs was collected based on their medical file.

#### 2.3.3. Quantification of SARS-CoV-2 S1RBD IgG Antibodies

To quantify S1RBD IgG, the DBS samples were analyzed using the SARS-CoV-2 S1RBD IgG ELISA assay (ImmunoDiagnostics Limited, Hong Kong) as previously described [19]. Briefly, a spot of 6 mm diameter was punched from the DBS and placed into 250 µL of 1x S1RBD IgG ELISA buffer. After a one-hour incubation at 37 °C, the extract was diluted 100-fold and transferred to a S1RBD-coated well plate. The remainder of the procedures followed the manufacturer’s ELISA assay instructions, with an optical density (OD) reading at 450 nm using the Behring ELISA Processor III (Siemens AG, Munich, Germany). Samples with OD higher than that of the highest standard value were retested using 10- and 1000-fold dilutions. Antibody concentrations (IU/mL) were calculated using a four-parameter logistic (4PL) regression curve (GraphPad Prism version 10.3.0 (GraphPad Software Inc., San Diego, CA, USA) based on a set of SARS-CoV-2 antibody standards provided in the kit. A seropositivity cutoff value of 26 IU/mL was experimentally determined [19].

#### 2.3.4. Detection of SARS-CoV-2 S1RBD IgG Antibodies

For detection of the presence of SARS-CoV-2 IgG (anti-spike) antibodies, point-of-care COVID-19 IgG/IgM rapid test cassettes (Healgen Scientific LLC, Houston, TX, USA) were performed using capillary blood obtained from the fingerprick, in SCOPE1 only. The results of these tests were used to classify antibody responders and non-responders to primary course vaccination (subset II).

#### 2.3.5. Statistical Analysis

The median age was calculated along with the interquartile range (IQR), while absolute numbers and proportions were reported for sex, infection history, and care dependency level. Infection-primed was defined as having a reported SARS-CoV-2 infection ≥ 14 days prior primary course vaccination. SARS-CoV-2 infections were reported through the online survey by a positive PCR and/or antigen test or COVID-19 confirmed by a CT scan. A breakthrough infection was defined as a reported positive PCR and/or antigen test or COVID-19 confirmed by a CT scan ≥ 14 days after primary course vaccination.

S1RBD IgG geometric mean antibody concentrations (GMC) were reported with 95% confidence intervals (CI), calculated with the log-normal method. First, data were log-transformed, then a mixed effects model was fitted and confidence intervals were computed in log scale (log_10_) and back-transformed into the original scale.

For all statistical testing, S1RBD IgG concentrations were log-transformed (log_10_) and normality was checked visually using QQ-plots. Data on epidemiological waves in Belgium with the respective dominant variant of concern (VOC) during the study follow-up were extracted from Sciensano [15,16]. A complete overview of the S1RBD IgG dynamics across the study was given descriptively (no statistical tests were applied on this data).

To compare S1RBD IgG antibody concentration over time after primary course and booster vaccination between infection-naive and infection-primed NHRs, subset I was used, but NHRs with a breakthrough infection (n = 61), those who received a first booster before sampling in October 2021 (8 months) (n = 49) and those who received a second booster during follow-up (n = 70), were excluded after that respective event. Based on these data, a mixed effects model with multiple comparisons was applied with timepoint, infection status, and the interaction term between timepoint and infection status as fixed effects and participants as random effect to account for within-participant correlation over the different timepoints. Comparisons between timepoints were performed on data from infection-naive and infection-primed NHRs combined between 2 versus 4, 6, 8, and 10 months, and 10 months versus 13, 16, and 19 months. Multiple unpaired t-tests were performed to assess differences in S1RBD IgG levels between infection-naive and infection-primed NHRs. All p-values were adjusted for multiple comparisons using a Bonferroni correction.

To assess the differences in S1RBD IgG levels after first booster vaccination between antibody responders and non-responders to primary course vaccination (subset II), a mixed effects model was applied with multiple comparisons. Timepoint, (non-)responder group, and the interaction term between timepoint and (non-)responder group were chosen as fixed effects and participant was chosen as random effect. Non-responders and responders were compared per timepoint and Bonferroni correction was applied to adjust for multiple comparisons. NHRs who remained seronegative after the first booster were excluded from this comparison for month 10.

To assess differences in S1RBD IgG levels between those who received a second booster and those who did not, a mixed effects model with multiple comparisons was applied with timepoint, vaccination status, and the interaction term of timepoint and vaccination status as fixed effects and participant as random effect on subset III. Multiple comparisons were performed between timepoints within a group and between groups within a timepoint, with all p-values adjusted for multiple comparisons using a Bonferroni correction.

The mixed effect models used a compound symmetry covariance matrix and the included variables were selected a priori. Missing completely at random (MCAR) was assumed. The within- and between-subject variances and missing observations per model are reported in Appendix A.

Adjusted two-sided *p*-values of ≤0.05 were considered statistically significant. All statistical analyses were performed using GraphPad Prism version 10.3.0 (GraphPad Software Inc., San Diego, CA, USA).

## 3. Results

### 3.1. Participant Characteristics

Participant characteristics for subsets I, II, and III are presented in Table 1. The median age among NHRs ranged between 86 and 89 years old across the three subsets. The majority of NHRs were female and were low care-dependent in all three subsets. In subset II, nearly all NHRs were infection-naive.

### 3.2. Dynamics of S1RBD IgG Levels Among Belgian NHRs Throughout Three Vaccination Rounds and Different Epidemiological Waves

Figure 2 gives an overview of the dynamics of S1RBD IgG levels among Belgian NHRs between February 2021 and September 2022, throughout three vaccination rounds and five epidemiological waves (subset I). As shown in Figure 2, S1RBD IgG levels were impacted by the booster vaccination rounds and/or the epidemiological waves driven by different VOCs. Additionally, S1RBD IgG levels varied widely among individuals, resulting in broader confidence intervals in some cases. Overall, higher and more durable S1RBD IgG levels were observed after the first booster vaccination compared to after primary course vaccination. S1RBD IgG concentrations after primary course and first booster dose without the effect of breakthrough infections are presented in the subsequent figure.

Among the 200 NHRs included in Figure 2, one breakthrough infection occurred between February and April 2021 (predominant VOC Alpha), 25 breakthrough infections occurred between December 2021 and March 2022 (predominant VOC Delta/Omicron BA.1), 29 breakthrough infections occurred between March and June 2022 (predominant VOC Omicron BA.2), and 6 breakthrough infections between June and September 2022 (VOC Omicron BA.5). Although S1RBD IgG antibody levels were higher during the Omicron waves compared to the Alpha and Delta waves, more breakthrough infections were reported during the Omicron waves.

### 3.3. S1RBD IgG Antibody Levels in Infection-Naive and Infection-Primed NHRs After Primary Course and First Booster Vaccination

Figure 3 compares S1RBD IgG antibody levels after primary course and first booster vaccination between infection-naive and infection-primed NHRs. After primary course vaccination, infection-naive NHRs had >10 times lower S1RBD IgG levels compared to infection-primed NHRs at all timepoints (*p* < 0.0001). Moreover, antibody levels gradually significantly decreased in the 8 months after primary course vaccination among NHRs from a GMC of 955 IU/mL (95% CI: 689–1323 IU/mL) to 240 IU/mL (95% CI: 157–365 IU/mL) (*p* < 0.0001) (infection-naive and infection-primed combined). Following the administration of a first booster dose, looking at infection-naive and infection-primed NHRs together, S1RBD IgG levels increased to a significantly higher level (3306 IU/mL, 95% CI 2622–4170) than what was observed directly after primary course vaccination (955 IU/mL, 689–1323 IU/mL) (*p* < 0.0001). This was mainly due to higher S1RBD IgG levels in infection-naive NHRs after a first booster dose, as here, ±10 times higher concentrations (2168 IU/mL, 95% CI 1554–3027 IU/mL) were observed compared to directly after primary course vaccination (292 IU/mL, 95% CI 197–432 IU/mL). Contrastingly, among infection-primed NHRs, S1RBD IgG levels after the first booster dose were comparable to what was observed directly after primary course vaccination (*p* = 0.483) (Appendix A).

The interaction term of infection status and timepoint was statistically significant (*p* < 0.0001), demonstrating different antibody dynamics in infection-naive versus infection-primed NHRs.

Yet, after first booster administration, differences in S1RBD IgG antibody levels between infection-naive and infection-primed NHRs were only significant in month 10 (*p* < 0.0001) and 13 (*p* < 0.001). In contrast to the period following primary course vaccination but preceding booster, where antibody levels gradually and significantly decreased among NHRs, S1RBD IgG antibody levels remained stable after the first booster dose (*p* > 0.05).

### 3.4. SARS-CoV-2 Antibody Response After First Booster in Primary Course Non-Responders Versus Responders

Figure 4 shows S1RBD IgG levels among non-responders versus responders to primary course vaccination. The majority of NHRs (56/61, 92%, and 95% CI: 82–97%) who did not develop an antibody response after primary course vaccination did have detectable S1RBD IgG levels after receiving a first booster dose. However, these S1RBD IgG levels after first booster administration were approximately 10 times lower among initial non-responders (594 IU/mL, 95% CI: 416–849 IU/mL) compared to responders (4642 IU/mL, 95% CI: 3577–6022 IU/mL) (*p* < 0.0001). The interaction term of the (non-)responder group and timepoint was statistically significant (*p* < 0.0001), demonstrating the different antibody dynamics between both groups. Among non-responders, 8% (5/61) still did not have detectable antibodies after the first booster dose (S1RBD IgG < 26 IU/mL).

### 3.5. S1RBD IgG Antibody Levels After Administration of Two Booster Doses Versus a Single Booster Dose in NHRs (Flanders vs. Brussels/Wallonia)

Figure 5 presents the S1RBD IgG levels in Flanders and Brussels/Wallonia before (June 2022) and after (September 2022) the second booster vaccination round took place in Flanders. NHRs in Flanders who received a second booster had significantly higher S1RBD IgG levels in the timepoint after (8447 IU/mL, 95% CI: 7171–9950 IU/mL) compared to before second booster administration (4538 IU/mL, 95% CI: 3697–5570 IU/mL) (*p* < 0.0001). S1RBD IgG levels among NHRs who did not receive a second booster dose (Brussels/Wallonia) remained stable in the timepoint after (*p* = 0.063). The interaction term of region and timepoint was statistically significant (*p* < 0.0001), demonstrating the different antibody dynamics between both groups.

Between June and September 2022, 5 (5/186, 3%) and 7 (7/237, 3%) breakthrough infections occurred among NHRs who received a second booster dose and NHRs who did not, respectively (excluded from analysis).

**Figure 5 vaccines-13-00409-f005:**
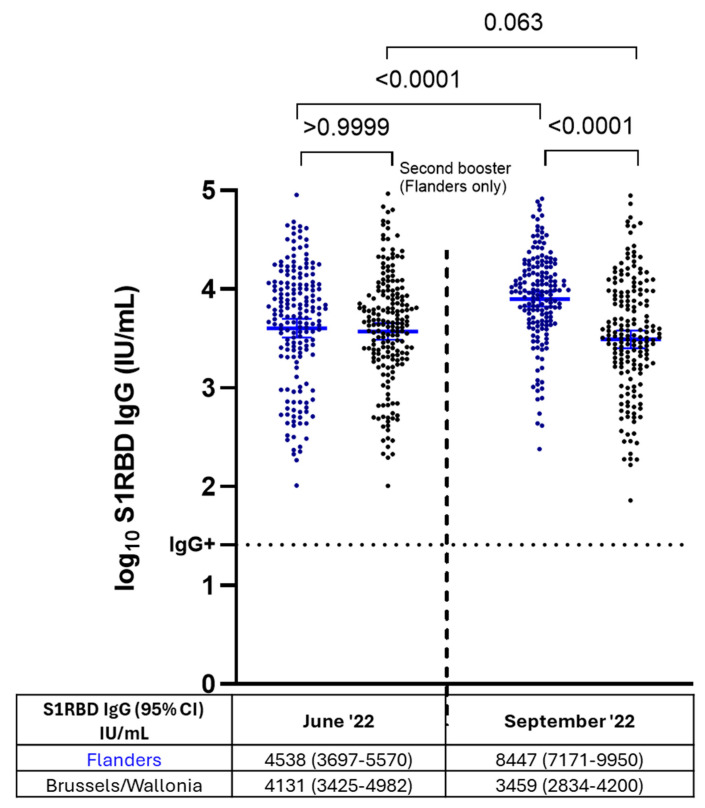
S1RBD IgG antibody levels before and after administration of two booster doses versus a single booster dose in nursing home residents (Flanders versus Brussels/Wallonia), n = 411. Blue: nursing home residents (NHRs) who received a second booster dose in the summer of 2022 (Flanders); N= 181, NHRs who received a second booster before sampling in June 2022 (n = 69) and NHRs who had a breakthrough infection between June and September 2022 (n = 5) were excluded. Black: NHRs who did not receive a second booster dose in the summer of 2022 (Brussels/Wallonia); N = 230 NHRs who had a breakthrough infection between June and September 2022 (n = 7) were excluded. S1RBD IgG geometric mean concentrations (GMC) in IU/mL with 95% CI are presented below each timepoint per group. Log_10_ GMCs are presented by horizontal lines and error bars. The horizontal dashed line represents the cutoff for SARS-CoV-2 seropositivity (log_10_ 26 IU/mL). CI: confidence interval. IU/mL: international units/mL.

## 4. Discussion

This observational serosurveillance study investigated S1RBD IgG antibody levels in Belgian NHRs across three vaccination rounds between February 2021 and September 2022. After primary course vaccination, antibody levels significantly declined over time, with infection-naive NHRs having concentrations over 10 times lower than infection-primed NHRs. The first booster elicited significantly higher antibody levels compared to primary course vaccination for infection-naive NHRs, while infection-primed NHR levels remained comparable. The vast majority (92%, 95% CI: 82–97%) of non-responders to primary course vaccination did develop antibodies after the first booster, though their response remained weaker (594 IU/mL, 95% CI: 416–849 IU/mL) than that of initial responders (4642 IU/mL, 95% CI: 3577–6022 IU/mL). After the second booster, significantly higher antibody concentrations were observed in NHRs from Flanders, while antibody levels remained stable in Brussels and Wallonia, where the second booster was not yet administered.

Over the 19-month follow-up period, antibody dynamics varied over time across different vaccination rounds and epidemiological waves. Although NHRs were vaccinated, some experienced a breakthrough infection. Most breakthrough infections in our study population were observed when the Omicron subvariants were predominant, despite booster vaccines being implemented and antibody levels being higher compared to the pre-Omicron period. In line with our findings, others demonstrated that breakthrough infections are more common for Omicron compared to the preceding Alpha and Delta variants, as primary course COVID-19 vaccination has been shown to be less protective to Omicron infection [20,21]. This suggests that breakthrough infections were driven more by variant immune escape than by insufficient antibody levels, yet disease severity of Omicron is shown to be lower compared to other variants [22,23].

Regarding the effect of vaccination, previous studies demonstrated that infection-naive NHRs exhibit significantly lower antibody responses to primary course vaccination compared to infection-primed NHRs, which is consistent with our findings [4,8,24,25,26]. Following the administration of the first booster, antibody levels among NHRs increased substantially due to the heightened responses in infection-naive individuals, while levels among infection-primed NHRs remained comparable to levels observed after primary course vaccination. As a result, the differences in antibody levels between infection-naive and infection-primed NHRs diminished, ultimately resulting in comparable levels. This finding highlights the critical role of booster vaccination in enabling infection-naive NHRs to develop a strong antibody response, comparable to that of infection-primed NHRs, who already reached such levels after primary course vaccination. Furthermore, antibody levels among NHRs remained stable following booster vaccination, in contrast to the decline observed after primary course vaccination. In line with our findings, a study among French NHRs found that anti-spike antibody levels remained high at least three months after first booster vaccination [27]. However, contrastingly, a study among NHRs in the UK showed a 21–78% decay in anti-spike antibody levels within 100 days after first booster vaccination [28]. Likewise, a study among NHRs in the US showed declining anti-spike antibody levels within six months after first booster vaccination [29]. Possibly, differences in vaccine brand could explain these conflicting findings, as among the latter two studies, also NHRs that received mRNA-1273 vaccines were included.

Additionally, we showed that the majority of antibody non-responders did present an antibody response after first booster vaccination. Nevertheless, this booster response was lower than the booster response among initial responders. These findings highlight that the immune response can be enhanced by re-introducing doses in these poor responders. Nevertheless, for the minority who does not benefit from repeated immunization, other strategies such as adapted vaccines or monoclonal antibody therapy should be explored. Others observed similar findings in immunocompromised individuals who did not show an antibody response after primary course vaccination, with some responding to booster vaccination while others did not [30,31,32,33]. A study by Alejo et al. in immunocompromised individuals showed that 50% of subjects that did not respond to a booster dose did respond to a second booster dose [34]. Supporting this, a systematic review by Martinelli et al. showed that a fourth dose was associated with improved seroconversion and antibody titer levels in immunocompromised individuals [35].

The regional differences in vaccination policy between Flanders (second booster) and Brussels/Wallonia (no second booster) allowed us to evaluate the impact of a second booster dose on antibody levels in nursing home residents and compare it to a control group (NHR). Antibody levels increased after the second booster, but remained stable over three months in NHRs who did not receive it. Consistent with our findings, another study reported a rise in anti-spike SARS-CoV-2 antibodies among NHRs following a second booster [36]. Additionally, McConeghy et al. demonstrated that a second booster provided better protection against severe COVID-19 outcomes compared to a single booster dose [37]. Nevertheless, during our follow-up (June–September 2022), breakthrough infections were rare, and infection rates were similar between NHRs with and without a second booster. Therefore, while our data suggest that the given second booster increased antibody levels among NHRs in Flanders, we could not assess any differences in protection between NHRs who did receive an early second booster and NHRs who did not.

Although this study is unique through its specific population and long follow-up time, there are also limitations. Namely, breakthrough infections and history of infection were based on the reporting of positive RT-PCR, antigen, and/or CT scan test results, therefore, underestimation of these breakthrough infections and infection history is likely due to asymptomatic infections or lack of reporting. Additionally, although we identified poor antibody responders and waning after COVID-19 vaccination, the translation of antibody levels into clinical protection against COVID-19 remains challenging, especially in light of different variants. As our assay detected binding antibodies against the wild-type SARS-CoV-2 variant, we were not able to assess variant-specific neutralization. Nevertheless, SARS-CoV-2 binding antibodies have previously been shown to be a correlate of protection [38,39,40], yet other immune factors, such as cellular immune responses, may also play a role in clinical protection [41,42]. Lastly, the applied mixed effects models used a compound symmetry covariance matrix, assuming equal correlations and variances across timepoints, possibly reducing statistical power.

By tracking antibody titers over time, we aimed to assess the durability of the humoral response in this high-risk population. We identified infection-naive NHRs as a poor-responding group, to which vaccine strategies should be tailored. As we showed the immunogenic potential of additional doses in non-responders to primary course vaccination, priority should be placed on ensuring repeated vaccination for poorly responding NHRs. Moreover, as a minority of NHRs still failed to mount robust responses, complementary strategies, such as monoclonal antibody therapies or next-generation vaccines targeting conserved viral epitopes, should be explored for this group. Regional differences in vaccination policy and their impact on antibody levels, as seen between Flanders and Brussels/Wallonia, indicate that timing and access to boosters significantly influence serological outcomes. Although we were not able to assess any clinical impact of this difference in policy, efforts should be made to safeguard equitable and timely vaccination campaigns across regions to ensure consistent protection for all NHRs. Overall, this research reinforces the importance of a dynamic, evidence-based approach to vaccination strategies in NHRs. By prioritizing tailored interventions, regular booster doses, and monitoring of immune responses, public health efforts can better safeguard this vulnerable population against SARS-CoV-2 and future emerging pathogens.

## 5. Conclusions

This study mapped SARS-CoV-2 antibody dynamics in NHRs across three vaccination rounds in Belgium. Antibody levels increased significantly after the first booster in infection-naive NHRs, who initially had weak responses. While antibodies among NHRs waned within six months after primary vaccination, they remained stable for up to nine months post-booster. Among primary vaccine non-responders, 92% developed antibodies after the first booster, though at much lower levels. The second booster in Flanders further increased antibody levels, while levels remained stable in Brussels and Wallonia, where it was not administered. These findings highlight the need for tailored vaccination strategies, especially for infection-naive NHRs. Repeated vaccination improves immune responses, but alternatives are needed for persistent non-responders. Regular immune monitoring is crucial to guide evidence-based vaccination strategies and protect vulnerable populations such as NHRs against SARS-CoV-2 and future infectious threats.

## Figures and Tables

**Figure 2 vaccines-13-00409-f002:**
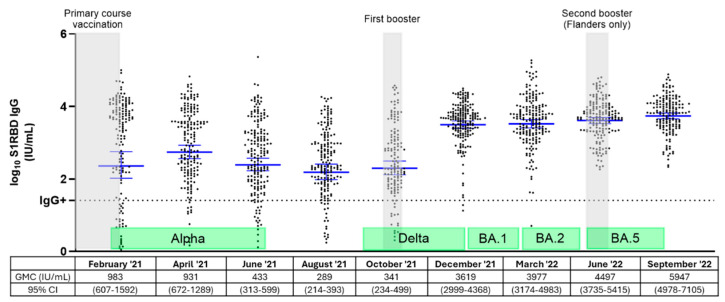
Overview of S1RBD IgG antibody levels among NHRs (n = 200) in Belgium from February 2021 to September 2022. S1RBD IgG geometric mean antibody concentrations (GMC) in IU/mL and 95% confidence intervals are presented below each timepoint and presented as log_10_ GMCs by blue lines with error bars. The horizontal dashed line represents the cutoff for SARS-CoV-2 seropositivity (log_10_ 26 IU/mL). Grey bars indicate the timing of the vaccination rounds in the nursing homes included in the subset. Green bars indicate epidemiological waves in Belgium with the respective dominant variant of concern [15,16]. IU/mL: international units/mL; CI: confidence interval. BA.1: Omicron BA.1; BA.2: Omicron BA.2; and BA.5: Omicron BA.5.

**Figure 3 vaccines-13-00409-f003:**
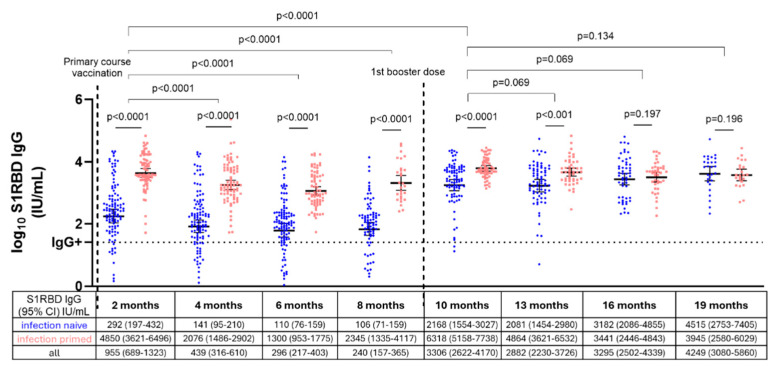
S1RBD IgG dynamics in infection-naive (n = 119) and infection-primed NHRs (n = 81) after primary course vaccination and first booster administration. Blue dots represent infection-naive NHRs, red dots represent infection-primed NHRs. S1RBD IgG geometric mean concentrations (GMC) in IU/mL with 95% CI are presented per timepoint for all subjects and separate for infection-naive/infection-primed subjects below the graph and log_10_ GMCs by horizontal lines and error bars for the latter. The mixed effects model and unpaired t-tests were applied. The first timepoint after vaccination was considered as baseline. The lower lines with p-values represent the comparison between infection-naive and infection-primed NHRs within one timepoint, while upper lines represent comparisons between timepoints for infection-naive and infection-primed NHRs combined. NHRs with a breakthrough infection and those who received a second booster were excluded at that respective timepoint. The horizontal dashed line represents the cutoff for SARS-CoV-2 seropositivity (log_10_ 26 IU/mL). IU/mL: international units/mL. CI: confidence interval.

**Figure 4 vaccines-13-00409-f004:**
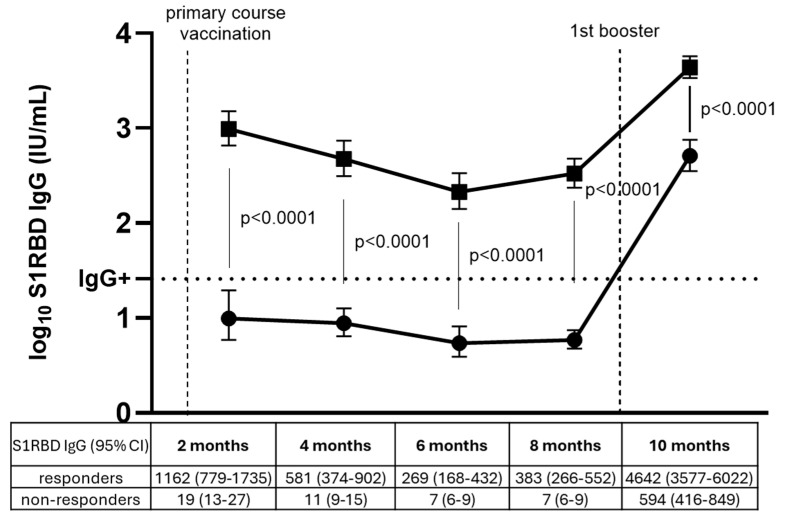
Geometric mean S1RBD IgG concentrations among responders (squares) (n = 60) and non-responders (dots) (n = 61) to primary course vaccination before and after first booster dose. Geometric mean S1RBD IgG concentrations (GMCs) in IU/mL with 95% confidence intervals are presented under each timepoint per group. Log_10_ GMCs are represented by dots among vaccine non-responders and squares among responders after primary course vaccination (month 2, 4, 6, and 8) and after the first booster dose (month 10) with error bars (95% CI). SARS-CoV-2 breakthrough cases were excluded from the analysis after the timepoint of infection (n = 23; 14 among non-responders, 9 among responders). Non-responders who remained seronegative after the first booster were excluded from this comparison for month 10. P-values are shown for the comparison between responders and non-responders per timepoint. The horizontal dashed line represents the cutoff for SARS-CoV-2 seropositivity (log_10_ 26 IU/mL). IU/mL: international units/mL. CI: confidence interval.

**Table 1 vaccines-13-00409-t001:** Participant characteristics in the NHR subsets I, II, and III.

		Subset I (n = 200)	Subset II (n = 121)	Subset III (n = 411)
Median age (IQR)		86 (81–91)	89 (84–92)	88 (81–91)
Female, n (%)		159 (80%)	96 (79%)	307 (75%)
Infection-primed, n (%)		81 (41%)	4 (3%)	137 (33%)
Care dependency level, n (%)	Low care dependency(category O, A, and B)	130 (65%)	79 (65%)	281 (68%)
	High care dependency(category C, Cd, and D)	70 (35%)	42 (35%)	128 (31%)

## Data Availability

The sponsor of this study (Sciensano) shares ownership of the data. Therefore, the data are available upon request and after permission of the sponsor.

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
