# Peer review of "Dynamics of SARS-CoV-2 IgG in Nursing Home Residents in Belgium Throughout Three BNT162b2 Vaccination Rounds: 19-Month Follow-Up"

_vaccines, 2025, doi:10.3390/vaccines13040409_

Round 1
Reviewer 1 Report
Comments and Suggestions for Authors
My comments to the authors are in the attached Word document.

Author Response
Reviewer 1
More information is needed on the mixed effects model used for statistical analysis to compare S1RBD IgG antibody concentration over time. Please revise. Three predictors were included in the model (study group, time (categorical) and the interaction between study group and time). Participant was included as a random effect in the model. Comment on or describe the variance-covariance structure (autoregressive, or compound symmetry) used for the model. Reporting the two variances (within-subject and between subject in the manuscript will be valuable to inform the design of a future study. Please discuss missing data. This model assumes the data is missing at random. The results and interpretation for the interaction test are not reported in the Results section. Were these analyses also adjusted for sex, site, and care dependency?
Figures 3 and 5 and the Supplemental file: Instead of using "ns," report the actual P-value, even if it's greater than .05. The actual P-values are important and not providing exact P-values is a form of incomplete reporting.
We would like to thank the reviewer for his/hers suggestions to improve our manuscript. We have extracted the following points from the text above. Our reply is marked in blue.
- Adding GMCs to abstract
We agree with the reviewer and have added GMCs with confidence intervals to the abstract to enhance its content. Additionally, we have added the GMCs throughout the main text of the results section.
- Methods: How were the 200 nursing home residents from Subset I randomly selected: a simple random sample or stratified random sample.
A simple random sample was taken within the SCOPE2 study population. The random sample was not stratified for infection status, we clarified this in the methods (line 88) and the supplementary file.
- Design: Power and sample size justification are not discussed, perhaps fixed according to the enrollment in the SCOPE study and study selection criteria.
We apologize for this missing information and have updated the revised manuscript with details on the sample size.
For subset I, sample size was chosen arbitrarily while for subset II and III, sample size was limited by the enrollment in the SCOPE study and the study selection criteria. A post-hoc sample size calculation was performed and showed an achieved power of 100% for all subsets for a repeated-measures ANOVA with within-between interaction, assuming an effect size of 0.25 (medium effect) and an alpha of 0.05 (G Power Version 3.1) [1].
We have added this to the manuscript in line 112-117.
[1] "G*Power 3: A flexible statistical power analysis program for the social, behavioral, and biomedical sciences," vol. 39, ed. US: Psychonomic Society, 2007, pp. 175-191.
- Analytic plan for the lognormal model.
We have added details on the computation of confidence intervals in line 177-180.
- Discuss the variance/covariance structure of the mixed effects models
The applied mixed models in our manuscript use a compound symmetry covariance matrix. This was added in line 217. We acknowledge that an autoregressive model might be more accurate to fit antibody dynamics over time, however, the analysis was performed in GraphPad Prism and the software does not allow for changing the variance structure (compound symmetry by default). We have included this in the limitation section, as this might impact statistical power (line 430-432).
- Report the between and within subject variability
We have included a table in the Supplementary Files (Supplementary Table 1) which includes the within and between subject variance for each mixed effects models. We refer to this table in line 220.
- Discuss missing data
We have added that missing of data was assumed to be random (line 219) and included the number of missing observations in Supplementary Table 1.
- Add results and interpretation for interaction test
For each mixed effects model, we have added the results of the interaction term in the results section (in text) in line 279, 308 and 331.
- Adjusted for sex , site and care dependency?
The variables included in the mixed effects model were selected a priori and did not include sex, site and care dependency.
One of our previous studies including data on antibody responses after primary course vaccination in nursing home residents demonstrated that sex is not associated with antibody levels in older adults [2]. Moreover, our study population has an overrepresentation of females (75% to 80% across the three subsets), resulting in too few participants per sex category. Secondly, although care dependency was found to be associated with antibody response after primary course vaccination in a previous study, we did not include it in this model to avoid overcomplication as the association was not that strong (with 95% CI of Cox Proportional Hazard leaning towards 1) [2]. Thirdly, nursing home was not included in the model since the number of participants per nursing homes was generally low (eg. median number of participants per NH=7, 30 NH for subset 1) and would lead to unstable estimates. Therefore, we opted for a model focusing on individual-level variation.
We added to the manuscript that variables were selected a priori (line 216).
[2] E. Meyers et al., "SARS-CoV-2 seroreversion and all-cause mortality in nursing home residents and staff post-primary course vaccination in Belgium between February and December 2021," Vaccine, vol. 51, p. 126865, 2025/04/02/ 2025, doi: https://doi.org/10.1016/j.vaccine.2025.126865.
- Add exact p-values
We agree that providing exact p-values is more transparent, we have adjusted this in the figures and the text.
- Mention sample sizes in the legend of Figure 3
We have added the sample size to the figure legend (line 289).
Reviewer 2 Report
Comments and Suggestions for Authors
This paper analyzes the serological data of hundreds of NHR in Belgium sampled from 2021 to 2022 to monitor the dynamics of their anti-SARS-CoV-2 spike RBD level. Althought a lot of similar conclusions were reported in other papers, it is still OK to publish it as it's a real-world experiment data and could arouse some readers' interest. Generally the paper is OK, with only minor issues in grammar/spelling and formatting.
Pay attention to these problems: log10 subscript is required for all
Figure legend is missing in Fig 4

Generally OK, but there are minor mistakes to correct. (see attachment)
Author Response
Reviewer 2
This paper analyzes the serological data of hundreds of NHR in Belgium sampled from 2021 to 2022 to monitor the dynamics of their anti-SARS-CoV-2 spike RBD level. Althought a lot of similar conclusions were reported in other papers, it is still OK to publish it as it's a real-world experiment data and could arouse some readers' interest. Generally the paper is OK, with only minor issues in grammar/spelling and formatting.
We confirm to have checked the marked version of the manuscript and have resolved the addtional minor comments. Our reply to your comments are marked in blue.
Pay attention to these problems: log10 subscript is required for all.
This was adjusted throughout the manuscript.
Figure legend is missing in Fig 4
We have checked this and all figures in the manuscript have a legend. The dots/square symbols are explained in the figure legend. We would like to ask if you could you inform us if there is still an issue here.
Consider mentioning limitation antibody assay only measured wild-type antibodies.
We agree that detecting binding antibodies against the wild-type variant does not provide information on variant-specific immunity. We have included this in the limitation section in line 422.
Reviewer 3 Report
Comments and Suggestions for Authors
1. Introduction
Lines 32–56 (Paragraph starting "The coronavirus disease...")
Strength: The authors provide a strong rationale for focusing on nursing home residents (NHR) and clearly establish the vulnerability of this population.
Suggestion:
Add a brief explanation of the term “infection-naive” vs “infection-primed” to assist readers unfamiliar with this immunological distinction. For instance:
"Infection-naive individuals are those with no prior confirmed SARS-CoV-2 infection, whereas infection-primed individuals have had confirmed infection prior to vaccination, potentially resulting in a more robust immune memory response."
Add this clarification around line 56, just after “...poor- or non-responders [4, 8].”
🔹 2. Methods – Study Design and Population
Lines 131–148 (Paragraph starting "The data in this publication were collected...")
Strength: The SCOPE1 and SCOPE2 study structure is described clearly.
Suggestion:
While the overview of Subsets I–III is well presented, key inclusion/exclusion criteria are only referenced as being in the Supplementary File.
To improve transparency, please add a concise bullet-style list of the main criteria for each subset directly in the main text (e.g., age cut-off, booster timing, exclusion after breakthrough infection). This will help readers understand cohort composition without needing to consult the supplement.
Recommend inserting at line 148, right after "More details on the subset selection and inclusion and exclusion criteria are found in the Supplementary File."
3. Methods – Questionnaires
Lines 189–197 (Paragraph starting “Sociodemographic and clinical data...”)
Issue: It's not entirely clear whether infection history data (e.g., infection-naive status) were validated through official records or self-/nurse-reported.
Suggestion:
Clarify the reliability of infection status data. For example:
“Infection history was reported by head nurses based on medical records and/or test confirmations (PCR/antigen), though retrospective misclassification of asymptomatic or unreported infections cannot be ruled out.”
Add at line 194, before the sentence “Moreover, during each round of sample collection...”
4. Methods – Antibody Quantification and Statistical Analysis
Lines 215–217 (Statistical analysis paragraph)
Issue: While it’s stated that log-transformed antibody levels were assumed to be normally distributed, it is unclear if normality was tested.
Suggestion:
State explicitly how normality was verified. If not tested, briefly explain reliance on literature precedent.
“Log-normality of antibody concentrations was assessed visually via Q-Q plots.”
—or—
“Normality was assumed based on prior serological studies using similar ELISA measurements in elderly populations.”
Insert around line 217.
5. Results – Non-Responders vs Responders
Lines 309–315 (Paragraph starting “Figure 4 shows S1RBD IgG levels among non-responders…”)
Strength: Excellent data presentation showing improvement in non-responders after first booster.
Suggestion:
While the 92% conversion is a powerful finding, you might add a sentence discussing whether the achieved titers (~594 IU/mL) among non-responders are likely to correlate with clinical protection — especially given recent studies on correlates of protection.
Add at line 315, after “…detectable antibodies after the first booster dose.”
“It remains uncertain whether these antibody levels confer sufficient clinical protection, particularly against variants with immune escape features such as Omicron.”
6. Discussion – Omicron Breakthroughs
Lines 393–405 (Paragraph starting “Over the 19-month follow-up period…”)
Issue: The phrase “despite higher antibody levels” may confuse readers if not contextualized.
Suggestion:
Clarify that Omicron’s immune escape capabilities, not a failure of antibodies per se, likely account for the higher breakthrough rate.
Add sentence around line 400:
“This suggests that breakthrough infections were driven more by variant immune escape than by insufficient antibody levels alone.”
7. Discussion – Booster Response in Non-Responders
Lines 435–455 (Paragraph starting “Additionally, we showed that the majority…” )
Suggestion:
You may strengthen this section by more directly tying the observed improvement in non-responders to potential policy recommendations — such as advocating for repeat boosters in this subgroup or personalized vaccine schedules.
Add to line 453:
“These results suggest that non-responders may benefit from personalized vaccine strategies involving early or repeated boosting.”
8. Language & Grammar Edits
Minor edits throughout could improve clarity. For example:
-
Line 85: "Regular immune monitoring is could aid..." → “Regular immune monitoring could aid...”
-
Line 344: “differences in S1RDB IgG antibody levels” → should be S1RBD (typo)
-
Line 661: “asses” → should be assess
Author Response
Reviewer 3 - Reply is marked in blue
- Introduction
Lines 32–56 (Paragraph starting "The coronavirus disease...")
Strength: The authors provide a strong rationale for focusing on nursing home residents (NHR) and clearly establish the vulnerability of this population.
Suggestion:
Add a brief explanation of the term “infection-naive” vs “infection-primed” to assist readers unfamiliar with this immunological distinction. For instance:
"Infection-naive individuals are those with no prior confirmed SARS-CoV-2 infection, whereas infection-primed individuals have had confirmed infection prior to vaccination, potentially resulting in a more robust immune memory response."
Add this clarification around line 56, just after “...poor- or non-responders [4, 8].”
We thank the reviewer for this suggestion. We have added an explanation on the terms infection-naive and infection-primed in line 95-97 to assist the reader. In line 164, a more detailed explanation is given on how being infection-primed was determined.
? 2. Methods – Study Design and Population
Lines 131–148 (Paragraph starting "The data in this publication were collected...")
Strength: The SCOPE1 and SCOPE2 study structure is described clearly.
Suggestion:
While the overview of Subsets I–III is well presented, key inclusion/exclusion criteria are only referenced as being in the Supplementary File.
To improve transparency, please add a concise bullet-style list of the main criteria for each subset directly in the main text (e.g., age cut-off, booster timing, exclusion after breakthrough infection). This will help readers understand cohort composition without needing to consult the supplement.
Recommend inserting at line 148, right after "More details on the subset selection and inclusion and exclusion criteria are found in the Supplementary File."
We thank the reviewer for this suggestion, however, we would like to note that the inclusion and exclusion criteria are already described in the main text (line 80-117). The supplementary file includes additional information on the rationale for these selections. To avoid confusion, we have changed the sentence ‘More details on the subset selection and inclusion and exclusion criteria can be found in the Supplementary File.’ to ‘More details on the subset selection can be found in the Supplementary File’.
- Methods – Questionnaires
Lines 189–197 (Paragraph starting “Sociodemographic and clinical data...”)
Issue: It's not entirely clear whether infection history data (e.g., infection-naive status) were validated through official records or self-/nurse-reported.
Suggestion:
Clarify the reliability of infection status data. For example:
“Infection history was reported by head nurses based on medical records and/or test confirmations (PCR/antigen), though retrospective misclassification of asymptomatic or unreported infections cannot be ruled out.”
Add at line 194, before the sentence “Moreover, during each round of sample collection...”
We agree with the reviewer that line 146-149 was ambiguous on the source of data collection. Indeed, these data were collected through the nursing home head nurses based on the medical file of NHR. This was clarified in line 146.
- Methods – Antibody Quantification and Statistical Analysis
Lines 215–217 (Statistical analysis paragraph)
Issue: While it’s stated that log-transformed antibody levels were assumed to be normally distributed, it is unclear if normality was tested.
Suggestion:
State explicitly how normality was verified. If not tested, briefly explain reliance on literature precedent.
“Log-normality of antibody concentrations was assessed visually via Q-Q plots.”
—or—
“Normality was assumed based on prior serological studies using similar ELISA measurements in elderly populations.”
Insert around line 217.
We tested normality visually based on QQ plots. This was added to line 183-184.
- Results – Non-Responders vs Responders
Lines 309–315 (Paragraph starting “Figure 4 shows S1RBD IgG levels among non-responders…”)
Strength: Excellent data presentation showing improvement in non-responders after first booster.
Suggestion:
While the 92% conversion is a powerful finding, you might add a sentence discussing whether the achieved titers (~594 IU/mL) among non-responders are likely to correlate with clinical protection — especially given recent studies on correlates of protection.
Add at line 315, after “…detectable antibodies after the first booster dose.”
“It remains uncertain whether these antibody levels confer sufficient clinical protection, particularly against variants with immune escape features such as Omicron.”
Indeed, we agree with the reviewer that the presence of antibodies do not necessarily correlate with clinical protection of SARS-CoV-2. This is discussed in the discussion section line 424 to 431.
- Discussion – Omicron Breakthroughs
Lines 393–405 (Paragraph starting “Over the 19-month follow-up period…”)
Issue: The phrase “despite higher antibody levels” may confuse readers if not contextualized.
Suggestion:
Clarify that Omicron’s immune escape capabilities, not a failure of antibodies per se, likely account for the higher breakthrough rate.
Add sentence around line 400:
“This suggests that breakthrough infections were driven more by variant immune escape than by insufficient antibody levels alone.”
We thank the reviewer for this insightful suggestion. This was added to line 371.
- Discussion – Booster Response in Non-Responders
Lines 435–455 (Paragraph starting “Additionally, we showed that the majority…” )
Suggestion:
You may strengthen this section by more directly tying the observed improvement in non-responders to potential policy recommendations — such as advocating for repeat boosters in this subgroup or personalized vaccine schedules.
Add to line 453:
“These results suggest that non-responders may benefit from personalized vaccine strategies involving early or repeated boosting.”
We thank the reviewer for this suggestion. We have moved this section up to line 396-398 so that is directly tied to our findings.
- Language & Grammar Edits
Minor edits throughout could improve clarity. For example:
- Line 85: "Regular immune monitoring is could aid..." → “Regular immune monitoring could aid...”
This was adjusted.
- Line 344: “differences in S1RDB IgG antibody levels” → should be S1RBD (typo)
This was adjusted.
- Line 661: “asses” → should be assess
This was adjusted.
Round 2
Reviewer 1 Report
Comments and Suggestions for Authors
The authors have addressed my concerns. This revision is now acceptable for publication.